# Soil Fungal Community Characteristics at Timberlines of Sejila Mountain in Southeast Tibet, China

**DOI:** 10.3390/jof9050596

**Published:** 2023-05-21

**Authors:** Fei Cheng, Mingman Li, Yihua Ren, Lei Hou, Tan Gao, Peng He, Xiangsheng Deng, Jie Lu

**Affiliations:** 1Institute of Tibet Plateau Ecology, Tibet Agricultural & Animal Husbandry University, Nyingchi 860000, China; feicheng@gxu.edu.cn (F.C.);; 2Key Laboratory of Forest Ecology in Tibet Plateau, Tibet Agricultural & Animal Husbandry University, Nyingchi 860000, China; 3Guangxi Key Laboratory of Forest Ecology and Conservation, Forestry College, Guangxi University, Nanning 530004, China

**Keywords:** soil fungal diversity, fungal community structure, functional composition, tree density, timberline ecotone

## Abstract

Soil fungal community characteristics of alpine timberlines are unclear. In this study, soil fungal communities in five vegetation zones across timberlines on the south and north slopes of Sejila Mountain in Tibet, China were investigated. The results show that the alpha diversity of soil fungi was not different between the north- and south-facing timberlines or among the five vegetation zones. *Archaeorhizomyces* (Ascomycota) was a dominant genus at the south-facing timberline, whereas the ectomycorrhizal genus *Russula* (Basidiomycota) decreased with decreasing *Abies georgei* coverage and density at the north-facing timberline. Saprotrophic soil fungi were dominant, but their relative abundance changed little among the vegetation zones at the south timberline, whereas ectomycorrhizal fungi decreased with tree hosts at the north timberline. Soil fungal community characteristics were related to coverage and density, soil pH and ammonium nitrogen at the north timberline, whereas they had no associations with the vegetation and soil factors at the south timberline. In conclusion, timberline and *A. georgei* presence exerted apparent influences on the soil fungal community structure and function in this study. The findings may enhance our understanding of the distribution of soil fungal communities at the timberlines of Sejila Mountain.

## 1. Introduction

Alpine timberline, an ecological transition zone from forest to alpine krummholz, is one of the most important climate-driven ecological boundaries [1]. In ecotones, mountain forests gradually transition to scrubland or grassland habitats with an increase in altitude. Global warming is conducive to the regeneration and growth of high-altitude forests and has led to the upward climb of timberlines [2,3]. Concomitantly, the tree size, density and coverage at timberlines also change [2,3].

At timberlines, tree distribution results in various habitat patches with different climate [4,5], hydrology [6], soil and biological characteristics [7], providing a diverse microenvironment available for soil microorganisms. Soil–vegetation interactions of timberlines are strongly closed to soil fungal taxa with diverse ecological functions [8]. Tree species may also mediate the soil fungal community [9]; for example, dominant tree species at timberlines are cold-resistant coniferous trees, most of which are ectomycorrhizal. Ectomycorrhizal fungi promote nutrient absorption [10,11], the stress resistance [12,13] of trees and the establishment of seedlings [14,15] at timberlines. Ectomycorrhizal fungi rely on their hosts to spread, and a decrease in their richness with a rise in elevation can be observed at timberlines [8], but the decreased species richness is not necessarily linked to the decreased ectomycorrhizal symbiosis; for example, there is no host effect on ectomycorrhizal fungal richness, and even an increase in species richness with a host decrease can also be observed [16]. Therefore, the ectomycorrhizal species richness is not contrary to the presented important role of ectomycorrhizal symbiosis for certain tree species. While the ectomycorrhizal abundance increases with the host density, the dominant role of certain ectomycorrhizal taxa may result in a decrease in ectomycorrhizal species richness. Certainly, other potential factors also act on the richness of ectomycorrhizal fungi, such as the soil C/N [16] or pH of the timberlines [17]. *Suillus*, *Tomentella* and *Cortinarius* of Basidiomycota and *Cenococcum* of Ascomycota are dominant genera of ectomycorrhizal fungi at several timberlines all over the world [16,18,19,20].

Saprophytic fungi are devoted to the decomposition of soil organic matter for promoting the soil nutrient cycle [21]. Soil saprotrophic fungi are usually more abundant under forests than shrubs at timberlines; for instance, Mucoromycota [22]. Saprotrophic fungi from Basidiomycota, Ascomycota and Zygomycota also occur frequently in soils from timberline forests, which are rich in wooden substrates [20]. These saprophytic fungi are believed to promote the turnover of soil organic matter in these harsh ecosystems together with ectomycorrhizal fungi [22].

At present, reports on the soil fungi of timberlines are very limited. We know that the changes in the soil fungal community occur along the elevation gradient due to the tree height [8], but tree species and obvious transitional differences in the number and spatial distribution of trees at different timberline zones with different vegetation characteristics also result in soil environmental heterogeneity [8,23]. For instance, in the Shenlongjia timberline ecotone, the shrubbery has a considerable amount of total and available phosphorus in the soil compared to the coniferous forest, but the soil pH and humidity levels are reversed [20]. Furthermore, in Changbai Mountains, the soil in the tree islands within the birch timberline ecotone has a significantly higher water content, total carbon and total nitrogen than open areas at equal elevation [23]. These soil heterogeneity effects related to timberline vegetation on the soil fungi community of timberlines need to be further explored.

Sejila Mountain is located in southeast Tibet, China, and is characterized by natural alpine timberlines (an altitude of over 4000 m). As the received solar radiation at the timberline on the south slope of Sejila Mountain is six times as much as that on the north slope, the heliophile *Sabina saltuaria* and shade-tolerant *Abies georgei* are the dominant tree species at the timberlines on the south and north slopes, respectively. Studies have confirmed the ectomycorrhizae associations between *A. georgei* and Basidiomycetes (e.g., Russulaceae, Cortinariaceae and Amanitaceae) [24,25], whereas no evidence for a symbiotic relationship between *S. saltuaria* or genus *Sabina* and fungi has been found so far, and even the distribution of soil saprotrophic fungi remains unknown at the timberlines of Sejila Mountain. It is urgent to note that the timberline in southeast Tibet has risen 69 m in the past 100 years [3]. Therefore, this research investigated the soil fungal community characteristics at the north- and south-facing timberlines of Sejila Mountain to describe soil fungal community characteristics at the timberlines and clarify the relationship between soil fungal communities and vegetation and soil factors. The fundings will provide important reference value for accurately understanding the role of soil fungi in the changing timberline ecosystems under climate change. Because dominant plants have a great impact on soil fungal community assembly [8,9], we assumed that the mycorrhizal fungi are mainly affected by host distribution, and therefore the diversity, composition and function of mycorrhizal fungi is expected to change with the decrease in mycorrhizal host species.

## 2. Materials and Methods

### 2.1. Study Site

This study was conducted at the Sejila Mountain (29°10′–30°15′ N, 93°12′–95°35′ E, 2200 m–5400 m a.s.l.), located in Nyingchi, Tibet, China [26]. The mountain comprises part of the dark coniferous forest in southeast Tibet. The Sejila Mountain is located in the transition zone between semi-humid and humid areas in southeast Tibet. The mountain extends from northwest to southeast, forming a large range of east–west slope. This region is characterized by sub-alpine temperate semi-humid climate, warm winter, cool summer and distinctive dry and wet seasons. The region has an annual average temperature of 2.0 °C–4.5 °C. The highest and lowest monthly average temperatures of the region are 11.1 °C (July) and −14.0 °C (January), respectively [27]. The region has an average annual relative humidity of 78.8% and average annual evaporation of 544 mm, accounting for 48.0% of the average annual precipitation (1134 mm). The precipitation in spring and summer accounts for 79.4% of the annual precipitation [27]. The soil in the region is mainly mountain brown soil and acid brown soil (pH = 4–6) [26].

The investigated timberlines of Sejila Mountain were divided into the timberline on the south slope (hereinafter referred to as south timberline) and the timberline on the north slope (hereinafter referred to as north timberline). In general, timberline ecotones consist of forest line, tree line and tree species line [28,29]. The forest line is the upper limit of the closed forest, with a canopy density ≤ 0.2. The tree line is the altitude at which the tree height is less than 5 m. The tree species line is the highest altitude of tree growth and comprises the upper limit for growth of isolated dwarf, curved trees. The closed forest is below the forest line and the scrubland is above the tree species line [30]. According to this, five vegetation zones along elevational gradients (<200 m) at the two timberlines were sampled: closed forest (F), forest line (FL), tree line (TL), tree species line (TSL) and shrub (S) [31]. In this study, the closed forests mainly consisted of *Sabina saltuaria* at the south timberline and *Abies georgei* at the north timberline. The alpine shrubs were dominated by *Rhododendron tanastylum*. This is a shrub widely spread under the tree canopies and the open areas of the timberlines, occurring together with *Rhododendron aganniphum*. The tree coverage and density of *S. saltuaria* and *A. georgei* significantly decreased from the closed forests to the shrubs, whereas the coverage of *R. tanastylum* increased. The geographical location of the study areas is shown in Figure 1. Information on the study sites is shown in Table 1.

### 2.2. Plot Setting and Vegetation Survey

In mid-August 2019, three 20 m × 20 m plots were set up at each of the five vegetation zones at each timberline. The replicate plots were placed at least 50 m apart perpendicular to the elevational gradient. A total of thirty plots were established in this study. Species and coverage of trees and shrubs in each plot were recorded. Tree individual numbers were also recorded.

### 2.3. Sample Collection

Soils were sampled in each plot in an S-shape manner to cover the entire plot [32]. After removing litter and organic layers, a soil drill (diameter, 50 mm) was used to collect 15 soil cores (0–20 cm) from each plot to increase sample representativeness [32]. The soil samples were sealed in sterilized bags, marked and transported to the laboratory. The soils were sieved through a 2 mm sieve to remove gravels and roots. The soils with same fresh weight were taken from the soil samples of the same plot and then evenly pooled to form 1 composite soil sample. A total of 30 composite soil samples (2 slope aspects × 5 vegetation zones × 3 repeat plots) were obtained. The composite samples were then divided into two parts: one part was stored at −80 °C for subsequent microbiological analysis and the other part was air-dried to analyze soil chemical characteristics. A volumetric soil sampler (100 cm^3^) was used to collect an extra soil sample from each plot at a depth of 10 cm to analyze the soil physical characteristics.

### 2.4. Determination of Soil Physical and Chemical Properties

Soil porosity, soil bulk density and field holding water capacity were analyzed using the volumetric soil samplers following the methods described by State Forestry Administration [33]. The air-dried soil samples were sieved using a 0.15 mm sieve and then used for the analysis of soil pH, soil organic carbon, total nitrogen, total phosphorus, ammonium nitrogen and available phosphorus. The determination of these soil chemical indexes followed Lu’s experimental manual [32]. An electrode method with a 1:5 soil:water ratio was used to determine soil pH using a pH meter (PHSJ-5, Leici, Shanghai, China). Soil organic carbon and total nitrogen contents were determined via the potassium dichromate oxidation and Kjeldahl methods. Total phosphorus and available phosphorus contents were determined via the molybdenum antimony ascorbic acid colorimetry method, whereas ammonium nitrogen content was determined via the indigo blue colorimetry method.

### 2.5. High-Throughput Sequencing of Soil Fungi

#### 2.5.1. Soil Fungal Genomic DNA Extraction and PCR Amplification

PowerSoil^®^ DNA Isolation Kit (MO BIO Laboratories, Carlsbad, CA, USA) was used to extract soil fungal genomic DNA from 0.5 g fresh soil sample following the manufacturer’s protocol. The quality of genomic DNA was determined via electrophoresis using 1.2% agarose gel. The fungal ITS1 region was amplified using the universal primers ITS1F (5′-CTTGGTCATTTAGAGGAAGTAA-3′, coverage 89.8%) [34,35] and ITS2R (5′-GCTGCGTTCTTCATCGATGC-3′, coverage 93.7%) [35,36]. The PCR reaction system contained 4 µL of 5 × Fastpfu buffer, 2 µL of 2.5 mmol/L dNTPs, 0.8 µL of forward primer (5 mol/L), 0.8 µL of reverse primer (5 mol/L), 0.4 µL FastPfu Polymerase, 0.2 µL BSA and 10 ng template DNA, and was then supplemented with ddH_2_O to 20 µL. The PCR reaction conditions were: pre-denaturation at 95 °C for 3 min, 35 cycles × (denaturation at 95 °C for 30 s, renaturation at 55 °C for 30 s, extension at 72 °C for 45 s), extension at 72 °C for 10 min and incubation at 10 °C before sequencing. Each sample was run in triplicates. The PCR amplicons of different soil samples were homogenized to 10 nmol/L and mixed in equal volume for library construction.

#### 2.5.2. Library Construction and Sequencing

The library was constructed using TruSeq^®^ DNA PCR-Free Sample Preparation Kit. The library was then quantified using qubit and Q-PCR methods. Library sequencing was conducted using Illumina NovaSeq 6000. Each sample was tagged based on the barcode sequence and PCR-amplified primer sequence. Reads of each sample were flashed (V1.2.11, http://ccb.jhu.edu/software/FLASH/ (accessed on 20 May 2023)) [37] and spliced after the interception of the barcode and primer sequence. The spliced sequence represented raw tags, which were filtered [38] to obtain high-quality tags (clean tags). Quality control for the tags was conducted using QIIME (V1.9.1, http://qiime.org/scripts/split_libraries_fastq.html (accessed on 20 May 2023)) [39]. The process was as follows: (1) tag interception: raw tags were cut from the first low-quality base site with continuous low-quality value (default quality threshold < 19) and the base number reached the set length (default length value, 3); (2) length filtering of tags: the data set of tags obtained by intercepting tags was further filtered to obtain a continuous high-quality base length less than 75% of the tags length. Tag sequences obtained after quality control were compared with the species annotation database to detect chimeric sequences (https://github.com/torognes/vsearch/ (accessed on 20 May 2023)) [40]. The final effective tags were obtained by removing chimeric sequences [41]. The details for specific data preprocessing and quality control are shown in Appendix A. The sequenced raw data were uploaded on the NCBI SRA (accession number; PRJNA785564) (https://www.ncbi.nlm.nih.gov/bioproject/PRJNA785564 (accessed on 20 May 2023)).

#### 2.5.3. OTU Clustering and Taxonomic Annotation

High-quality sequences were clustered using UPARSE (V11, http://www.drive5.com/uparse/ (accessed on 20 May 2023)) [42] into OTUs (operational taxonomic units) with 97% similarity threshold. The sequence with the highest frequency in an OTU was selected as the representative sequence of that OTU. The blast method in QIIME (V1.9.1) (http://qiime.org/scripts/assign_taxonomy.html (accessed on 20 May 2023)) [39] and UNITE (V8.3) databases (https://unite.ut.ee/ (accessed on 20 May 2023)) [43] were used to annotate OTUs taxonomically (kingdom, phylum, class, order, family and genus, species). Multiple sequence alignment was conducted using MUSCLE (V5, http://www.drive5.com/muscle/ (accessed on 20 May 2023)) [44] to explore the phylogenetic relationship among all OTU representative sequences. The data for each sample were rarefied to the number of sequence reads obtained in the sample with the least amount of data (48,579 reads). After this, all the samples had saturating species accumulation curves (Appendix A), indicating that the sequencing depth was sufficient and that fungal diversity in the soil samples was effectively covered by sequencing. Thus, subsequent fungal diversity and community structure analyses were conducted based on the rarefied data. The workflow for sequencing, quality control and OTU annotation is shown in Figure 2. FUNGuild is a functional annotation tool used to predict the functional composition of soil fungal communities of fungal amplicons obtained through high-throughput sequencing and other methods. However, it can only annotate information of fungal species on trophic-mode levels (pathotroph, symbiotroph and saprotroph). The modes can be further subdivided into multiple guides. Each guide comprises the species with similar absorption and utilization of environmental resources [45]. Herein, the ecological function of soil fungi was described based on the mode and guild levels.

### 2.6. Data Analysis

In order to determine differences between the two timberlines in soil fungi, the north and south timberlines were indicated by “1” and “2”, respectively. Rarefaction curves were generated using R (V3.6.3) [46]. Venn diagrams were drawn using the R and VennDiagram (V1.7.1) package to show the quantitative distribution of soil fungal OTUs of different timberlines and vegetation zones [47]. The data were tested for homogeneity of variance before alpha diversity analysis. Levene’s test was conducted using R and car (V3.0.12) to check whether the data variances of timberlines and vegetation zones were equal (*p* > 0.05) [48]. If necessary, the data were log-transformed to obtain equal variances. Two-way ANOVA was used to test differences in vegetation and soil factors, soil fungal diversity indices between the two timberlines and the five vegetation zones using R. The tested alpha diversity indices were: observed species, Shannon entropy of counts (Shannon), Simpson’s index (Simpson), Chao1 richness estimator (Chao1), abundance-based coverage estimator (ACE), Faith’s phylogenetic diversity metric (PD) and Good’s coverage of counts (Goods_coverage). An alpha diversity package of the scikit-bio development team (http://scikit-bio.org/docs/latest/generated/skbio.diversity.alpha.html#module-skbio.diversity.alpha (accessed on 20 May 2023)) was used to calculate the indices. The data are expressed as mean ± standard deviation (SD).

The R and ggplot2 (V3.3.5) package [49] were used to show the accumulative bar diagrams, representing the relative abundance of soil fungal phyla and predicted fungal modes of each vegetation zone. Heat maps were generated using the R and pheatmap (V1.0.12) package [50] to show the relative abundance of the top 10 fungal genera and fungal guilds of each vegetation zone. The sample differences in species complexity were evaluated using beta diversity analysis. Beta diversity of the weighted UniFrac distance was calculated using QIIME (V1.9.1). PERMANOVA tests (permutation = 999) based on Bray–Curtis distance were conducted using the R and vegan (V2.5.7) package to determine the soil fungal community structure differences [51]. The similarity in soil fungal community structure of different timberlines and vegetation zones was analyzed using a principal coordinate analysis (PCoA) in R and ggplot2 package.

The relationships among fungal, vegetative and edaphic factors were assessed using redundancy analysis (RDA) and Pearson correlation. The data were standardized using z-score for data uniformity before RDA analysis. Multicollinearity factors (variance inflation factor > 10) for vegetation and soil were excluded from analysis. RDA (permutation = 999) was conducted based on the standardized data. Canoco (V5.0) was used for RDA analysis [52]. Pearson correlation was performed in R (V3.6.3) [46].

## 3. Results

### 3.1. Vegetation and Soil Characteristics

#### 3.1.1. Vegetation

The two-way ANOVA (Table 2) showed that tree density (F = 5.21, *p* = 0.03) at the north timberline was significantly higher than that at the south timberline. Shrub coverage (F = 11.72, *p* < 0.01) in the shrublands surpassed that in the forests and forest lines. Moreover, tree coverage (F = 111.73, *p* < 0.01) and density (F = 25.14, *p* < 0.01) were considerably greater in the forests and lower in the shrublands compared to the other four vegetation zones, respectively. The interaction between the timberline site and vegetation zone had a significant effect on shrub coverage (F = 3.18, *p* = 0.04). Specifically, at the south timberline, shrub coverage in the forest was lower than that in the other vegetation zones, and increased gradually from forest to shrubland. Tree coverage and density, however, were highest in the forest compared to other zones, exhibiting a decline from forest to shrubland. The trend on the north timberline was similar, but shrub coverage in the shrubland was significantly higher than that in the forest and forest line. Nonetheless, tree coverage and density were substantially greater in the forest than in the other vegetation zones.

#### 3.1.2. Soil

The two-way ANOVA (Table 3) indicated that the soil field water-holding capacity (F = 6.14, *p* = 0.02) at the north timberline was significantly higher than that at the south timberline, while the soil bulk density (F = 3.16, *p* = 0.04) in the forests and tree lines was significantly higher than that in the tree species lines. There was also a significant interaction between the timberline site and vegetation zone regarding soil bulk density (F = 4.00, *p* = 0.02). Specifically, the soil bulk density in the forest at the south timberline displayed a notably higher value than that in other vegetation zones, while the soil field water-holding capacity in the tree species line surpassed both the forest and shrubland by a significant margin. At the north timberline, only the tree line’s soil bulk density displayed a significant elevation from other vegetation zones.

The soil pH (F = 45.21, *p* < 0.01) and available phosphorus content (F = 27.80, *p* < 0.01) were significantly higher at the south timberline when compared to the north timberline, whereas the soil ammonium nitrogen content (F = 13.84, *p* < 0.01) was significantly higher at the north timberline (Table 3). The available phosphorus content (F = 3.56, *p* = 0.02) was significantly higher in the forests than in the other three vegetation zones, except for tree lines. Moreover, the interaction between the timberline site and vegetation zone had a significant effect on soil pH (F = 16.30, *p* < 0.01), total phosphorus (F = 5.04, *p* = 0.01) and ammonium nitrogen (F = 3.83, *p* = 0.02). The pH levels in the south timberline’s forest and forest line were notably distinct from other vegetation zones, while the soil organic carbon concentration in the shrubland was considerably higher than that in the forest. Furthermore, the total phosphorus content of the tree line was significantly greater than shrub. On the other hand, the north timberline exhibited lower soil pH and available phosphorus levels in the forest compared to other vegetation zones. Additionally, the forest and tree line had a significantly higher soil ammonium nitrogen content than the tree species line and shrub.

The soil C/N (F = 87.27, *p* < 0.01) was significantly higher at the north timberline than at the south timberline, while shrublands had a significantly higher C/N (F = 36.02, *p* < 0.01) than the other vegetation zones (Table 3). Additionally, soil C/N (F = 17.68, *p* < 0.01) and C/P (F = 3.63, *p* = 0.02) were significantly affected by the interaction between the timberline site and vegetation zone. Notably, the C/N and C/P of shrub at the south timberline were significantly higher than those of other vegetation zones, while the N/P of shrub was significantly higher than that of the forest and tree line. Finally, the C/N of shrub at the north timberline was significantly higher than that of other vegetation zones, except for the forest line.

### 3.2. Overview of High-Throughput Sequencing

A total of 2,718,759 raw paired-end (PE) reads were obtained from all samples. Samples had 77,454–99,866 reads (average: 90,625). A total of 1,952,609 effective reads were obtained after filtering steps. The samples had average effective reads of 65,087 (range: 49,412–69,415).

### 3.3. Soil Fungal Diversity

Alpha diversity indices were not significantly different between the timberlines and vegetation zones, and there was no interaction between these factors (F = 1.333–3.568, *p* = 0.073–0.262, Table 4, Appendix A).

### 3.4. Soil Fungal Community Composition

#### 3.4.1. Relative Abundance of Soil Fungal Phyla and Genera

Soil fungi were classified into 12 phyla, 43 classes, 104 orders, 201 families, 343 genera and 311 species. Notably, soil fungal communities of the two timberlines were dominated by Basidiomycota (12–80%) and Ascomycota (10–57%) (Figure 3). An obvious change in relative abundance from Basidiomycota to Ascomycota was observed with higher elevation at the north timberline. The relative abundance of Basidiomycota significantly decreased from forest (71%) to shrub (19%). On the contrary, the relative abundance of Ascomycota increased from forest (16%) to shrub (53%) (Figure 3). The relative abundance of Ascomycota was higher compared with that of Basidiomycota at all vegetation zones of the south timberline. Across all vegetation zones, the relative abundance of Mortierellomycota was also rather high (7–25%), especially at the south timberline.

The soil fungal communities at the two timberlines comprised different dominant fungal genera (Figure 4). *Archaeorhizomyces* (8–30%) of Ascomycota was the most dominant genus across all vegetation zones and was more abundant in the south timberline ecotone. The relative abundance of *Russula* of Basidiomycota ranged from 8% to 30% at the north timberline (except in the shrub zone (0.43%)), but only from 0.18% to 0.34% at the south timberline.

#### 3.4.2. Soil Fungal Community Similarity

The PERMANOVA showed significant differences in soil fungal community composition (based on OTUs) between the two timberlines (R^2^ = 0.109, F = 3.441, *p* = 0.001) and among the vegetation zones (R^2^ = 0.082, F = 2.488, *p* = 0.002). These differences were verified through the PCoA explaining 41.22% of the total variation (Figure 5). The soil fungal communities were distinguished by the PC1 axis (27.23%) based on timberline (Figure 5). The fungal community compositions in the closed forest, forest line and tree line at the north timberline (all with a dominance of *A. georgei*) were similar, clustering at the left of the axis, whereas fungal communities at the south timberline were more similar to vegetation types with few or no *A. georgei* (tree species line and shrub) at the north timberline, clustering at the right of the axis. PC2 (13.99%) indicated the difference in fungal communities across the vegetation zones at the southern timberline. The fungal communities of forest and shrub at the south timberline were similar and were mainly distributed in the lower half of PC2, the fungal communities at the tree species line clustered in the middle and similar communities from other vegetation zones at the south timberline were scattered across PC2. However, the soil fungal communities of the forest, forest line and tree line at the north timberline mainly clustered in the middle of PC2.

### 3.5. Ecological Function of Soil Fungi

The Saprotroph mode (25–36%) and Soil_Saprotroph guild (16–30%) dominated at the south timberline, with little variation across the vegetation zones, whereas the Symbiotroph mode and Ectomycorrhizal guild were characteristic of the north timberline together with the Saprotroph mode and Soil_Saprotroph guild (Figure 6 and Figure 7). The relative abundance of the Symbiotroph mode (56–3%) and Ectomycorrhizal guild (41–1%) decreased across the vegetation zones at the north timberline, whereas the relative abundance of the Saprotroph mode (11–35%) and Soil_Saprotroph guild (8–30%) increased across the vegetation zones (Figure 6 and Figure 7).

### 3.6. Relationship between Soil Fungal Community Characteristics and Vegetation and Soil Factors

The first two axes of the RDA explain 25.95% of the total variation (16.80% for the first axis and 9.15% for the second axis) (Figure 8 and Table 5). The observed species richness, Chao1, ACE and PD were negatively correlated with the vegetation zone (altitude), shrub coverage, soil field holding-water capacity, ammonium nitrogen and C/N, but were positively associated with the soil pH and bulk density. Negative correlations of the timberline site with the Shannon and Simpson indexes were observed. Ascomycetes, saprophytic and soil saprophytic fungi were negatively correlated with the timberline site, tree factors and soil ammonium nitrogen, and positively correlated with the soil pH, whereas the opposite was the case for Basidiomycetes, symbiotic and ectomycorrhizal fungi.

However, Pearson correlation analysis further showed that the observed species richness, PD and Shannon and Simpson indexes were significantly positively associated with soil pH, but the Shannon and Simpson indexes were obviously negatively associated with soil ammonium nitrogen (Figure 9C,F). At the north timberline, the coverage and density of trees were significantly negatively related to Ascomycetes, saprophytic and soil saprophytic fungi, and significantly positively related to Basidiomycetes, symbiotic and ectomycorrhizal fungi (Figure 9A). The shrub coverage had contrary linkages compared with the tree factors. Therefore, the composition and function of soil fungal communities at the north timberline changed with the tree and shrub coverage and tree density. At the north timberline, the soil C/N and pH were negatively correlated with Basidiomycetes, symbiotic and ectomycorrhizal fungi, whereas ammonium nitrogen was positively correlated; the opposite was the case for Ascomycetes, saprophytic and soil saprophytic fungi, especially regarding soil C/N (Figure 9D). Unlike the north timberline, there was no significant correlation between soil fungi, vegetation and soil factors at the south timberline (Figure 9B,E). The Pearson correlation also indicated that the Chytridiomycota relative abundance significantly negatively responded to shrub coverage, soil field water-holding capacity, C/N and ammonium nitrogen, but significantly positively responded to soil bulk density and pH (Figure 8 and Figure 9F).

## 4. Discussion

The findings of the present study show that the composition and function of soil fungal communities between the north and south timberlines of Sejila Mountain had a strong site effect. The soil fungal community characteristics at the north timberline regularly changed with the tree coverage and density, but the tree species line and shrub were more similar to the vegetation zones of the south timberline regarding their traits. This indicates that the *A. georgei* presence also had an impact on the soil fungal community composition and function.

### 4.1. Distribution of Mycorrhizal Fungi and Its Relationship with Vegetation

The distribution of ectomycorrhizal fungi is strongly influenced by the tree species and their distribution [53]. In this study, *A. georgei* is a prominent species at the northern timberline ecosystem due to its preference for shade and moisture [54]. The ectomycorrhizal fungi predominantly occurred at the northern timberline and their abundance had a positive correlation with the density and coverage of *A. georgei*. The family Russulaceae is the dominant group of ectomycorrhizal fungi. *Russula* is the type genus of Russulaceae under Basidiomycetes. Its members can form ectomycorrhizae with trees of *Larix*, *Picea*, *Pinus*, *Fagaceae*, *Salicaceae* and *Tiliaceae* [55]. A study on Russulaceae fungi along the altitude gradient of 2600–4500 m in Sejila Mountain also showed that 70% of Russulaceae species are distributed under *A. georgei* forest, but they rarely appear in the high-altitude Rhododendron shrubs [24,25]. Several research studies have confirmed that *Russula* can form an ectomycorrhizal symbiosis with tree species of the *Abies* genus, such as *Abies lasiocarpa* [56], *Abies balsamea* [57], *Abies koreana* [58] and *Abies firma* [59]. This supports the notion that *A. georgei* is their potential host too. The number of *A. georgei* individuals has a great effect on the root density in soil, further affecting the relative abundance of ectomycorrhizal fungi [60]. This is the reason for the positive synergistic change in the relative abundance of Basidiomycetes, *Russula*, symbiotic and ectomycorrhizal groups with the tree coverage and density of *A. georgei*. This is basically consistent with our hypothesis of the host effect. Four species in *Russula* genus were identified in this study, but their relative abundances were extremely low. This implies that the high relative abundance of *Russula* was due to the presence of other unidentified species. The significantly negative correlation between the Shannon index and tree density at the north timberline indicates that the ectomycorrhizal fungi may inhibit other groups of fungi at the timberline. Owing to their remarkable cold tolerance, ectomycorrhizal fungi are considered to be crucial for seedling establishment, nutrient absorption and the distribution expansion of *A. georgei* [61]. In summary, the presence of *A. georgei* led to systematic changes in the soil fungal community characteristics.

### 4.2. Distribution of Saprophytic Fungi and Its Relationship with Vegetation

Soil saprotrophic fungal communities at the timberlines of Sejila Mountains are largely populated by the *Archaeorhizomyces* fungi. In this study, regarding the symbiosis relationship between *S. saltuaria* and mycorrhizal fungi, which has not been reported previously, here, it was also found that the low relative abundance of mycorrhizal fungi at the south timberline was dominated by *S. saltuaria*. However, there were plenty of *Archaeorhizomyces* belonging to the class Archaeorhizomycetes of Ascomycetes at the south timberline [62]. It is not clear enough whether the taxonomic status and ecological function of *Archaeorhizomyces* are a result of difficulties in isolation and culture. Only two species have been systematically described: *Archaeorhizomyces finlayi* [63] and *Archaeorhizomyces borealis* [62]. *Archaeorhizomyces* are non-mycorrhizal rhizosphere-associated fungi with saprotrophic activity. For example, in culture, *A. finlayi* grows slowly on both glucose and cellulose as a sole carbon source, indicating that it may be involved in decomposition [63]. At alpine timberlines and in other stressful ecosystems [22,64,65,66], *Archaeorhizomyces* is a common fungal group, similar to our study. *Archaeorhizomyces* also often dominate soils of rainforest [67], temperature forests [68], boreal forests [69] and arctic tundra [70]. This explains the strongly positive associations among Ascomycetes, *Archaeorhizomyces*, saprophytic and soil saprophytic groups in this study (Appendix A).

At the north timberline, *Archaeorhizomyces* were negatively correlated with Basidiomycetes, *Russula*, symbiotic and ectomycorrhizal groups and *A. georgei* coverage and density, but positively correlated with shrub coverage (Appendix A). This may be related to the alleviative competition with *Russula* with the decrease in *A. georgei* hosts at the north timberline. Although the precise ecological niches of *Archaeorhizomyces* remain unknown [63], the higher relative abundance of *Archaeorhizomyces* within the ecotone implies that fungi of *Archaeorhizomyces* are more suitable for this transitional habitat lacking tree cover. One species of *Archaeorhizomyces* was identified in the current study, and the relative abundance was low. This also indicates that the high relative abundance of *Archaeorhizomyces* is attributed to the presence of other unidentified species. Overall, it is highly probable that *Archaeorhizomyces* collaborate with ectomycorrhizal fungi to facilitate the circulation of materials and energy at the timberlines of Sejila Mountain.

### 4.3. Relationship between Fungal Distribution and Soil Factors

In this study, timberlines on different slopes and diverse vegetation zones at varying altitudes show considerable variations in both physical and chemical properties of the soil [71,72]. These variations play a significant role in shaping the traits of soil fungal communities, together with timberline vegetation [73]. At the north timberline, due to a deficiency in base cations in the litters, *A. georgei* litters may lead to soil acidification [74], resulting in a significant negative correlation between soil pH and tree density, coverage and ectomycorrhizal fungi. Previous studies have confirmed that ammonium nitrogen promotes the symbiosis between ectomycorrhizal fungi and trees [75,76,77]. The soil total nitrogen pool at the vegetation zones with more trees at the north timberline ensured a continuous supply of ammonium nitrogen. A decrease in the relative abundance of ectomycorrhizae at the tree species line and shrub may be related to a significant reduction in N pool supply, in addition to a reduction in host density. The soil stoichiometric ratios indicate the mineralization capacity for soil carbon, nitrogen and phosphorus [78,79]. The soil C/N at the north timberline increased with a decrease in tree coverage and density, whereas N/P decreased. This indicates that soil nitrogen tended to be mineralized at the vegetation zones with more trees, which is conducive for the accumulation of soil ammonium nitrogen [80]. On the contrary, soil nitrogen mineralization weakened at the tree species line and shrub, and ammonium nitrogen accumulation was low. It is noteworthy that, at the south timberline, there is no obvious correlation between soil fungal community characteristics and vegetation and soil factors. We speculate that there may be other potential factors that play an important role in regulating soil fungal communities, which needs further investigation.

## 5. Conclusions

In the present study, the soil fungal diversity, community structure and ecological function of the north and south timberlines of the Sejila Mountain in China’s southeastern Tibet along a 200 m elevation gradient were explored. The findings show that the site effect of timberlines and *A. georgei* presence exerted apparent influences on the soil fungal community structure and function at the two timberlines. These soil fungal community traits were related to coverage and density, soil pH and ammonium nitrogen at the north timberline, whereas they had no associations with the vegetation and soil factors at the south timberline.

## Figures and Tables

**Figure 1 jof-09-00596-f001:**
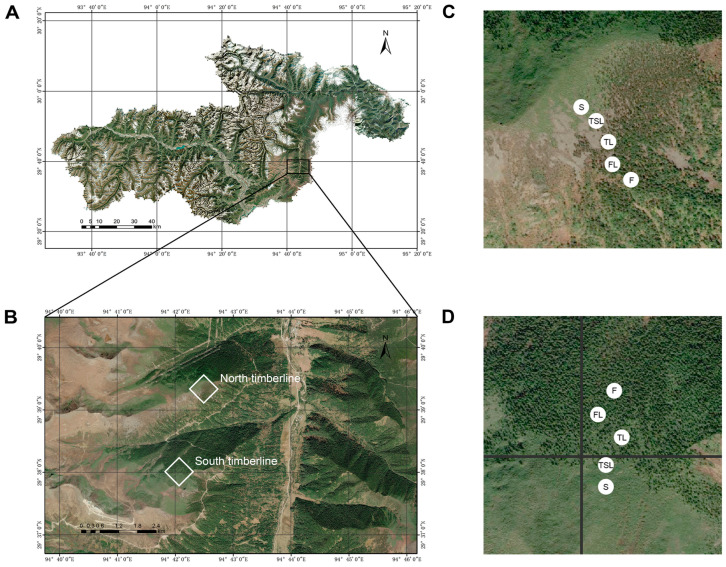
Geographical location of study areas. (**A**) Nyingchi County, Tibet, China, (**B**) timberline sites, (**C**) vegetation zones at the south timberline, (**D**) vegetation zones at north timberline. F, forest; FL, forest line; TL, tree line; TSL, tree species line; S, shrub.

**Figure 2 jof-09-00596-f002:**
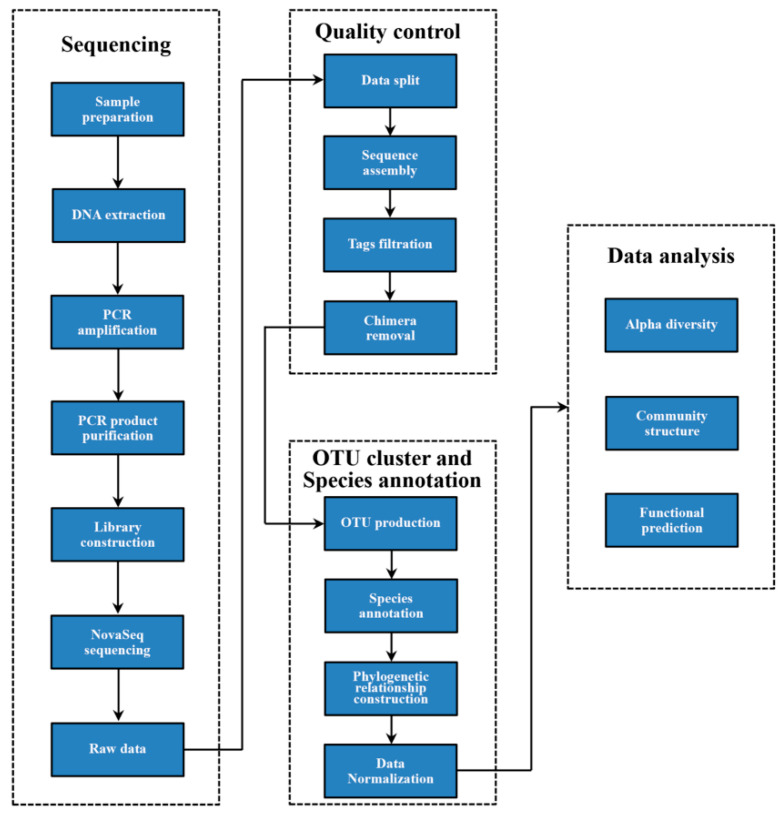
Workflow for sequencing and data analysis.

**Figure 3 jof-09-00596-f003:**
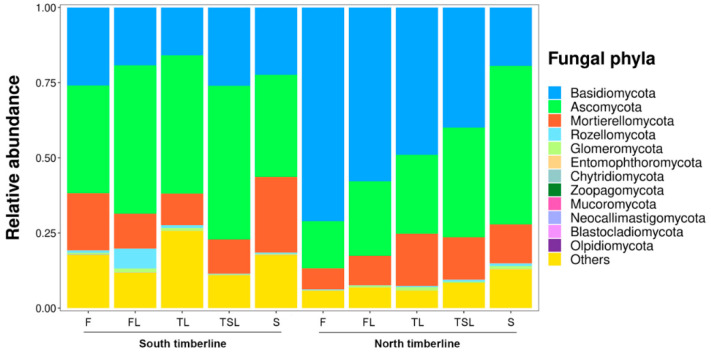
Accumulative bar diagram for relative abundance of soil fungal phyla at the south and north timberlines of Sejila Mountain. F, forest; FL, forest line; TL, tree line; TSL, tree species line; S, shrub.

**Figure 4 jof-09-00596-f004:**
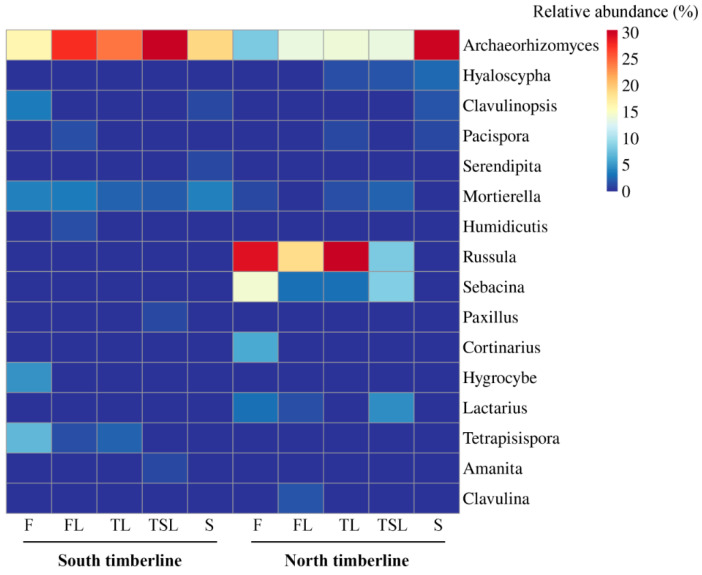
Heatmap for relative abundance of the top 10 fungal genera at the south and north timberlines of Sejila Mountain. F, forest; FL, forest line; TL, tree line; TSL, tree species line; S, shrub.

**Figure 5 jof-09-00596-f005:**
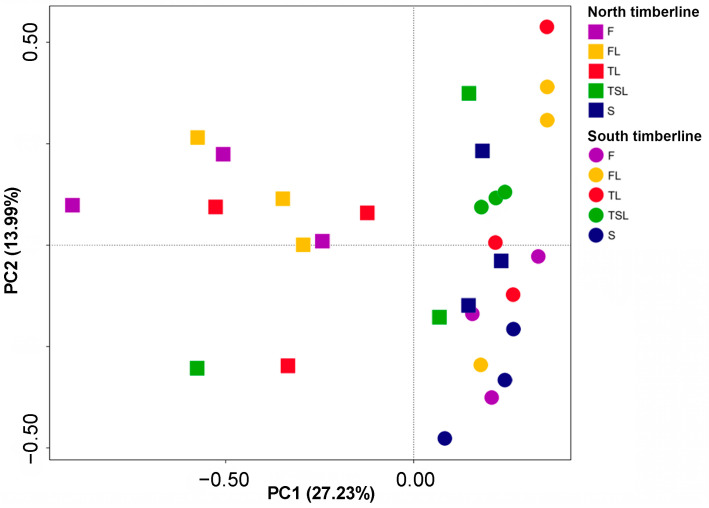
PCoA for dissimilarity in soil fungal communities at the south and north timberlines of Sejila Mountain. F, forest; FL, forest line; TL, tree line; TSL, tree species line; S, shrub.

**Figure 6 jof-09-00596-f006:**
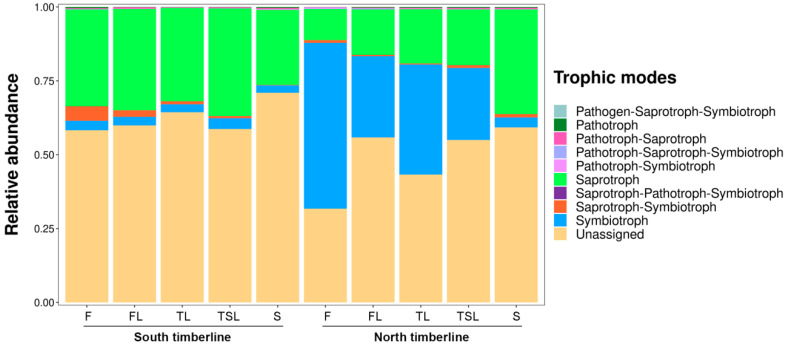
Accumulative bar diagram for relative abundance of soil fungal modes at the south and north timberlines of Sejila Mountain. F, forest; FL, forest line; TL, tree line; TSL, tree species line; S, shrub.

**Figure 7 jof-09-00596-f007:**
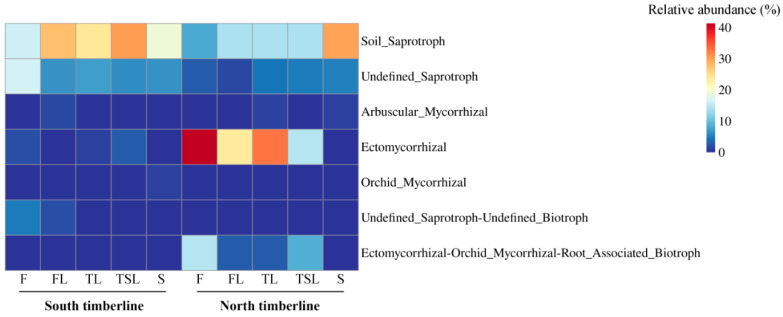
Heatmap for relative abundance of soil fungal guilds at the south and north timberlines of Sejila Mountain. F, forest; FL, forest line; TL, tree line; TSL, tree species line; S, shrub.

**Figure 8 jof-09-00596-f008:**
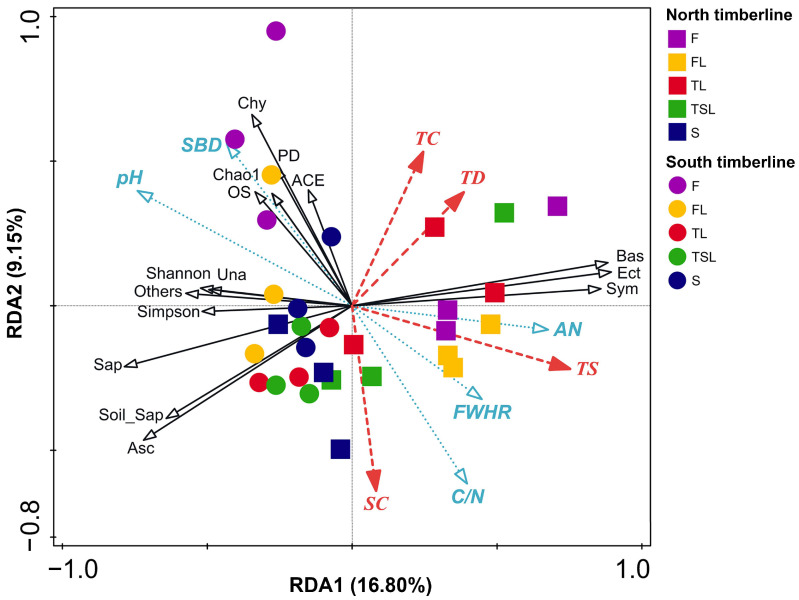
RDA for relationship between soil fungal diversity, community structure, mode, guild and vegetation and soil factors at the south and north timberlines of Sejila Mountain. F, forest; FL, forest line; TL, tree line; TSL, tree species line; S, shrub; TC, tree coverage; TD, tree density; TS, timberline site; SC, shrub coverage; SBD, soil bulk density; FWHC, field water-holding capacity; AN, ammonium nitrogen; C/N, carbon-to-nitrogen ratio of soil; OS, observed species; Shannon, Shannon entropy of counts; Simpson, Simpson’s index; Chao1, Chao1 richness estimator; ACE, abundance-based coverage estimator; PD, Faith’s phylogenetic diversity metric; Bas, Basidiomycota; Asc, Ascomycota; Chy, Chytridiomycota; Sap, Saprotroph; Sym, Symbiotroph; Una, unassigned; Sap-Sym, Saprotroph–Symbiotroph; Soil_Sap, Soil_Saprotroph; Ect, Ectomycorrhizal.

**Figure 9 jof-09-00596-f009:**
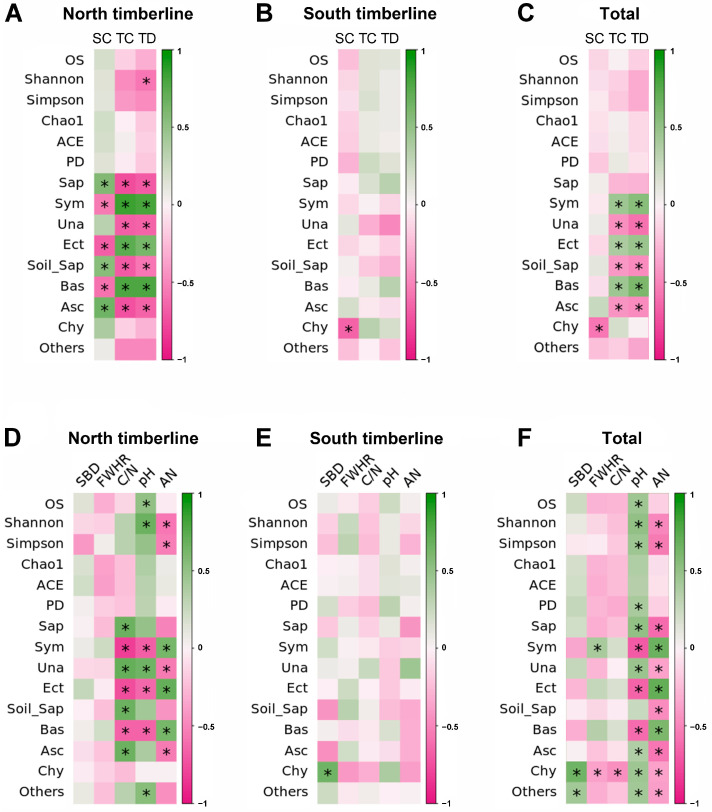
Pearson correlation between soil fungal community characteristics and vegetation and soil factors at the timberlines of Sejila Mountain. Correlations between soil fungal diversity and vegetation at (**A**) the north timberline, (**B**) the south timberline and (**C**) the two timberlines. Correlations between soil fungal community composition, function and vegetation at (**D**) the north timberline, (**E**) the south timberline and (**F**) the two timberlines. The significance level is shown by * (*p* < 0.05). TC, tree coverage; TD, tree density; TS, timberline site; SC, shrub coverage; SBD, soil bulk density; FWHC, field water-holding capacity; AN, ammonium nitrogen; C/N, carbon-to-nitrogen ratio of soil; OS, observed species; Shannon, Shannon entropy of counts; Simpson, Simpson’s index; Chao1, Chao1 richness estimator; ACE, abundance-based coverage estimator; PD, Faith’s phylogenetic diversity metric; Bas, Basidiomycota; Asc, Ascomycota; Chy, Chytridiomycota; Sap, Saprotroph; Sym, Symbiotroph; Una, unassigned; Sap-Sym, Saprotroph–Symbiotroph; Soil_Sap, Soil_Saprotroph; Ect, Ectomycorrhizal.

**Table 1 jof-09-00596-t001:** Environmental characteristics of study sites.

Timberline Sites ^a^	Vegetation Zones	Altitude (m)	Slope Degree (°)	Dominant Vegetation
South	F	4280–4287	18~28	*S. saltuaria*, *R. aganniphum*, *R. nyingchiens*
	FL	4335–4360	11~17	*S. saltuaria*, *R. aganniphum*, *R. nyingchiens*
	TL	4354–4380	10~25	*R. aganniphum*, *R. nyingchiense*, *S. saltuaria*
	TSL	4384–4390	6~18	*R. aganniphum*, *R. nyingchiense*
	S	4390–4416	1~7	*R. aganniphum*
North	F	4223–4250	18–23	*A. georgei*, *R. aganniphum*, *R. nyingchiense*
	FL	4321–4339	8~18	*A. georgei*, *R. aganniphum*, *R. nyingchiense*
	TL	4353–4370	11~15	*R. aganniphum*, *R. nyingchiense*, *A. georgei*
	TSL	4387–4392	13~18	*R. aganniphum*, *R. nyingchiense*
	S	4403–4412	12~21	*R. aganniphum*

^a^ F, forest; FL, forest line; TL, tree line; TSL, tree species line; S, shrub; *S. saltuaria*, *Sabina saltuaria*; *A. georgei*, *Abies georgei*; *R. aganniphum*, *Rhododendron aganniphum*; *R. nyingchiense*, *Rhododendron tanastylum*.

**Table 2 jof-09-00596-t002:** Vegetation factors at the south and north timberlines of Sejila Mountain.

Timberline Sites	Vegetation Zones	Vegetation Factors ^a^		
		SC (%)	TC (%)	TD (Trees/hm^2^)
South	F	56.67 ± 16.07 b	65.00 ± 5.00 a	341.67 ± 160.73 a
	FL	81.67 ± 2.89 a	46.67 ± 7.64 b	175.00 ± 66.14 b
	TL	85.00 ± 5.00 a	30.00 ± 5.00 c	58.33 ± 14.43 bc
	TSL	91.67 ± 5.77 a	13.33 ± 2.89 d	41.67 ± 14.43 bc
	S	91.67 ± 5.77 a	0.00 ± 0.00 e	0.00 ± 0.00 c
	Mean	81.33 ± 15.17	31.00 ± 24.29	123.33 ± 144.07
North	F	78.33 ± 7.64 b	65.00 ± 13.23 a	400.00 ± 125.00 a
	FL	80.00 ± 5.00 b	43.33 ± 5.77 b	308.33 ± 62.92 a
	TL	86.67 ± 2.89 ab	36.67 ± 2.89 b	166.67 ± 62.92 b
	TSL	86.67 ± 7.64 ab	18.33 ± 2.89 c	50.00 ± 25.00 bc
	S	95.00 ± 0.00 a	0.00 ± 0.00 d	0.00 ± 0.00 c
	Mean	85.33 ± 7.67	32.67 ± 23.59	185.00 ± 167.12
F_Timberline_		2.36	0.61	5.21
P_Timberline_		0.14	0.44	0.03
F_Zone_		11.72	111.73	25.14
P_Zone_		0.00	0.00	0.00
F_Timberline×Zone_		3.18	0.73	0.96
P_Timberline×Zone_		0.04	0.58	0.45

^a^ Lowercase letters indicate significant differences between different vegetation zones at the same timberline. SC, shrub coverage; TC, tree coverage; TD, tree density; F, forest; FL, forest line; TL, tree line; TSL, tree species line; S, shrub.

**Table 3 jof-09-00596-t003:** Soil factors at the south and north timberlines of Sejila Mountain.

Timberline Sites	Vegetation Zones	Soil Factors ^a^											
		SP	SBD	FWHC	pH	SOC	TP	TN	AN	AP	C/N	C/P	N/P
		(%)	(g/cm^3^)	(%)		(g/kg)	(g/kg)	(g/kg)	(mg/kg)	(mg/kg)			
South	F	60.00 ± 2.91 a	0.97 ± 0.21 a	55.44 ± 11.05 b	5.58 ± 0.02 a	72.98 ± 17.77 b	0.75 ± 0.03 ab	3.40 ± 0.65 a	47.50 ± 9.66 a	2.14 ± 0.04 a	21.27 ± 1.41 c	97.11 ± 21.52 b	4.54 ± 0.79 b
	FL	98.15 ± 54.11 a	0.49 ± 0.28 b	102.55 ± 30.52 ab	5.36 ± 0.17 a	104.58 ± 17.03 ab	0.75 ± 0.10 ab	5.00 ± 1.40 a	60.08 ± 1.91 a	1.79 ± 0.14 a	21.38 ± 2.55 c	139.39 ± 6.40 b	6.61 ± 1.16 ab
	TL	73.16 ± 2.94 a	0.52 ± 0.24 b	120.83 ± 62.71 ab	5.02 ± 0.18 b	131.09 ± 29.51 ab	0.81 ± 0.06 a	4.99 ± 1.17 a	77.50 ± 20.91 a	2.74 ± 1.08 a	26.31 ± 0.28 b	162.51 ± 40.89 b	6.19 ± 1.62 b
	TSL	74.62 ± 2.95 a	0.36 ± 0.15 b	138.39 ± 31.62 a	4.90 ± 0.17 b	138.54 ± 30.23 ab	0.66 ± 0.05 bc	5.02 ± 0.99 a	68.25 ± 12.73 a	1.97 ± 0.48 a	27.55 ± 0.68 b	211.13 ± 40.73 ab	7.64 ± 1.29 ab
	S	74.33 ± 5.32 a	0.54 ± 0.06 b	109.52 ± 4.82 ab	4.86 ± 0.05 b	182.22 ± 79.28 a	0.57 ± 0.06 c	5.85 ± 2.45 a	76.75 ± 43.37 a	1.54 ± 1.57 a	30.99 ± 0.62 a	319.26 ± 125.41 a	10.26 ± 3.86 a
	Mean	76.05 ± 24.26	0.58 ± 0.27	105.35 ± 41.02	4.86 ± 0.31	125.88 ± 51.54	0.71 ± 0.10	4.85 ± 1.49	66.02 ± 22.43	2.03 ± 0.85	25.50 ± 4.04	185.88 ± 94.93	7.05 ± 2.62
North	F	65.43 ± 4.36 a	0.23 ± 0.10 b	165.69 ± 87.40 a	4.26 ± 0.15 b	139.47 ± 14.82 a	0.62 ± 0.06 a	5.20 ± 0.54 a	127.08 ± 14.13 a	2.22 ± 0.35 a	26.83 ± 0.50 d	223.37 ± 12.49 a	8.33 ± 0.59 a
	FL	67.53 ± 3.71 a	0.26 ± 0.16 b	168.39 ± 58.68 a	4.71 ± 0.19 a	161.81 ± 96.66 a	0.60 ± 0.06 a	5.32 ± 3.12 a	106.25 ± 46.90 ab	0.35 ± 0.15 b	30.26 ± 0.43 ab	262.84 ± 131.61 a	8.65 ± 4.23 a
	TL	67.12 ± 7.82 a	0.53 ± 0.22 a	96.90 ± 18.80 a	4.93 ± 0.25 a	154.84 ± 35.05 a	0.70 ± 0.06 a	5.45 ± 1.15 a	119.75 ± 15.08 a	0.52 ± 0.23 b	28.36 ± 0.43 c	221.52 ± 49.41 a	7.80 ± 1.61 a
	TSL	73.79 ± 8.50 a	0.20 ± 0.05 b	184.94 ± 70.43 a	4.82 ± 0.14 a	126.46 ± 8.77 a	0.65 ± 0.06 a	4.31 ± 0.23 a	63.25 ± 6.95 b	0.50 ± 0.08 b	29.33 ± 1.04 bc	194.46 ± 24.18 a	6.64 ± 0.87 a
	S	63.69 ± 7.35 a	0.29 ± 0.10 ab	130.75 ± 34.92 a	4.89 ± 0.26 a	151.23 ± 46.05 a	0.71 ± 0.10 a	4.92 ± 1.51 a	70.25 ± 7.65 b	0.40 ± 0.15 b	30.72 ± 0.20 a	209.58 ± 39.71 a	6.82 ± 1.31 a
	Mean	67.51 ± 6.63	0.30 ± 0.17	149.33 ± 59.89	4.72 ± 0.30	151.23 ± 46.05	0.66 ± 0.08	5.04 ± 1.46	97.32 ± 33.33	0.80 ± 0.76	29.10 ± 1.53	222.35 ± 60.89	7.65 ± 2.00
F_Timberline_		1.71	19.20	6.14	45.21	1.51	3.83	0.11	13.84	27.8	87.27	2.4	0.6
P_Timberline_		0.21	2.88	0.02	0.00	0.23	0.06	0.74	0.00	0.00	0.00	0.14	0.45
F_Zone_		1.03	3.16	1.22	1.05	1.31	2.71	0.50	1.81	3.56	36.02	2.06	0.85
P_Zone_		0.41	0.04	0.33	0.40	0.30	0.06	0.74	0.17	0.02	0.00	0.12	0.51
F_Timberline×Zone_		0.88	4.00	1.59	16.3	1.24	5.04	0.73	3.83	2.57	17.68	3.63	2.71
P_Timberline×Zone_		0.49	0.02	0.22	0.00	0.32	0.01	0.58	0.02	0.07	0.00	0.02	0.06

^a^ Lowercase letters indicate significant differences between different vegetation zones at the same timberline. F, forest; FL, forest line; TL, tree line; TSL, tree species line; S, shrub; SC, shrub coverage; TC, tree coverage; TD, tree density; SP, soil porosity; SBD, soil bulk density; FWHC, field water-holding capacity; SOC, soil organic carbon; TN, total nitrogen; TP, total phosphorus; AN, ammonium nitrogen; AP, available phosphorus; C/N, carbon-to-nitrogen ratio; C/P, carbon-to-phosphorus ratio; N/P, nitrogen-to-phosphorus ratio.

**Table 4 jof-09-00596-t004:** Soil fungal diversity at the south and north timberlines of Sejila Mountain.

Timberline Sites	Vegetation Zones	OS	ACE	Chao1	Shannon	Simpson	Goods_Coverage	PD
South	F	744 ± 109 a	870.534 ± 122.140 a	831.879 ± 125.155 a	5.665 ± 0.570 a	0.943 ± 0.023 a	0.997 ± 0.001 a	209.136 ± 33.325 a
	FL	665 ± 121 a	800.677 ± 123.711 a	764.611 ± 121.938 a	5.413 ± 0.472 a	0.943 ± 0.015 a	0.997 ± 0.001 a	181.342 ± 30.852 a
	TL	676 ± 204 a	807.454 ± 230.070 a	766.311 ± 231.254 a	5.425 ± 1.257 a	0.929 ± 0.066 a	0.997 ± 0.001 a	186.111 ± 66.182 a
	TSL	657 ± 52 a	790.033 ± 65.099 a	759.933 ± 70.602 a	5.676 ± 0.326 a	0.953 ± 0.011 a	0.997 ± 0.001 a	179.771 ± 14.494 a
	S	714 ± 32 a	867.160 ± 47.436 a	823.505 ± 33.026 a	5.307 ± 0.510 a	0.922 ± 0.044 a	0.996 ± 0.001 a	188.148 ± 18.947 a
	Mean	691 ± 107	827.172 ± 118.663	789.248 ± 118.057	5.497 ± 0.616	0.938 ± 0.034	0.997 ± 0.001	188.902 ± 33.486
North	F	603 ± 46 a	751.066 ± 124.208 a	718.568 ± 105.106 a	4.542 ± 0.130 a	0.869 ± 0.019 a	0.997 ± 0.001 a	167.133 ± 27.693 a
	FL	586 ± 70 a	693.782 ± 95.361 a	660.492 ± 76.691 a	5.000 ± 0.799 a	0.916 ± 0.062 a	0.997 ± 0.001 a	156.946 ± 31.501 a
	TL	712 ± 153 a	871.145 ± 173.517 a	823.665 ± 169.359 a	4.983 ± 0.976 a	0.877 ± 0.108 a	0.997 ± 0.001 a	190.444 ± 47.168 a
	TSL	609 ± 22 a	722.291 ± 32.316 a	694.143 ± 18.849 a	5.173 ± 0.545 a	0.922 ± 0.034 a	0.997 ± 0.000 a	165.631 ± 7.974 a
	S	655 ± 111 a	767.896 ± 121.008 a	739.167 ± 124.540 a	5.456 ± 0.416 a	0.941 ± 0.019 a	0.997 ± 0.000 a	172.730 ± 35.279 a
	Mean	633 ± 92	761.236 ± 118.176	727.207 ± 109.594	5.031 ± 0.626	0.905 ± 0.057	0.997 ± 0.001	170.577 ± 29.819
F_Timberline_		2.229	2.038	1.905	3.568	3.304	1.333	2.047
P_Timberline_		0.151	0.169	0.183	0.073	0.084	0.262	0.168
F_Zone_		0.499	0.610	0.518	0.237	0.619	0.292	0.373
P_Zone_		0.736	0.660	0.723	0.914	0.654	0.880	0.825
F_Timberline × Zone_		0.532	0.527	0.474	0.668	0.725	1.125	0.347
P_Timberline × Zone_		0.714	0.717	0.754	0.622	0.585	0.373	0.843

^a^ Lowercase letters indicate significant differences between different vegetation zones at the same timberline. F, forest; FL, forest line; TL, tree line; TSL, tree species line; S, shrub; OS, observed species; Shannon, Shannon entropy of counts; Simpson, Simpson’s index; Chao1, Chao1 richness estimator; ACE, abundance-based coverage estimator; Goods_coverage, Good’s coverage of counts; PD, Faith’s phylogenetic diversity metric; Timberline, timberline site; Zone, vegetation zone.

**Table 5 jof-09-00596-t005:** RDA result for vegetation, soil and fungi factors at the south and north timberlines of Sejila Mountain.

Factors ^a^	RDA1	RDA2	r^2^	*p*
SC	0.143641	−0.98963	0.496922	0.001
TC	0.351797	0.936076	0.436142	0.001
TD	0.632203	0.774803	0.357184	0.002
TS	0.966325	−0.25733	0.619121	0.001
SBD	−0.61957	0.784943	0.520066	0.001
FWHC	0.796237	−0.60498	0.319073	0.006
pH	−0.88802	0.459812	0.716007	0.001
C/N	0.531154	−0.84728	0.603805	0.001
AN	0.998002	−0.06318	0.470154	0.001
OS	−0.69521	0.718802	0.236768	0.025
Shannon	−0.99978	0.020968	0.26829	0.015
Simpson	−0.99411	−0.10838	0.265634	0.016
ACE	−0.62356	0.781778	0.199583	0.045
Chao1	−0.62749	0.778628	0.199686	0.043
PD	−0.52077	0.853696	0.27333	0.016
Asc	−0.83238	−0.55421	0.727328	0.001
Bas	0.98076	0.195217	0.808387	0.001
Chy	−0.49419	0.869353	0.54539	0.002
Others	−0.99535	0.096355	0.33698	0.005
Sap	−0.96414	−0.26541	0.659629	0.001
Sym	0.99291	0.118865	0.754456	0.001
Soil_Sap	−0.8569	−0.51549	0.548047	0.001
Ect	0.982745	0.184964	0.835384	0.001
Una	−0.99994	−0.0111	0.251452	0.022

^a^ SC, shrub coverage; TC, tree coverage; TD, tree density; TS, timberline site; SBD, soil bulk density; FWHC, field water-holding capacity; AN, ammonium nitrogen; C/N, carbon-to-nitrogen ratio of soil; OS, observed species; Shannon, Shannon entropy of counts; Simpson, Simpson’s index; Chao1, Chao1 richness estimator; ACE, abundance-based coverage estimator; PD, Faith’s phylogenetic diversity metric; Asc, Ascomycota; Bas, Basidiomycota; Chy, Chytridiomycota; Sap, Saprotroph; Sym, Symbiotroph; Soil_Sap, Soil_Saprotroph; Ect, Ectomycorrhizal; Una, unassigned.

## Data Availability

The sequenced raw data were uploaded on the NCBI SRA with an accession number PRJNA785564.

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
