# Peer review of "Soil Fungal Community Characteristics at Timberlines of Sejila Mountain in Southeast Tibet, China"

_jof, 2023, doi:10.3390/jof9050596_

Round 1
Reviewer 1 Report
The study by Cheng et al. has provided a very comprehensive insight into the soil fungal communities at timberlines of the Sejila Mountain in southeastern Tibet. The experimental design is presented in detail. However, the discussion needs to be expanded to include more literature. In its present state, it is presented more as a free interpretation of the results. The conclusion needs to be more soundly stated and not repeat results.
Table 3. should be presented in the Results section
Page 7, lines 34-36: move to the Results section
Author Response
Response to Reviewer 1
The study by Cheng et al. has provided a very comprehensive insight into the soil fungal communities at timberlines of the Sejila Mountain in southeastern Tibet. The experimental design is presented in detail.
However, the discussion needs to be expanded to include more literature. In its present state, it is presented more as a free interpretation of the results.
Response:
Thank you for taking the time to provide us with your valuable feedback. To further support the discussion section, we have added the following 14 references to support the discussion section in our revised manuscript. Please refer to the discussion section and references in the revised manuscript.
The conclusion needs to be more soundly stated and not repeat results.
Response:
Thank you very much for your comment. We have changed the sentences “In the present study, soil fungal diversity, community structure and ecological function of the north and south timberlines of the Sejila Mountain in China’s southeastern Tibet along a 200m elevation gradient were explored. The findings showed that the alpha diversity of soil fungi had no difference between the two timberline sites or among the five landscape sites. Archaeorhizomyces of Ascomycota was dominant at the south timberline, while Russula of Basidiomycota decreased with decreasing Abies georgei coverage and density at the north timberline. The soil saprotroph-relating fungi were dominant, but changed little among the landscape sites at the south timberline, while ectomycorrhizal fungi decreased with the hosts at the north timberline. The site effect of timberlines and A. georgei presence exerted apparent influences on the soil fungal community structure and function at the two timberlines. These community traits were related to coverage and density, soil pH and ammonium nitrogen at the north timberline, while they had no associations with the vegetation and soil factors at the south timberline. These findings may enhance our understanding of the distribution of soil fungal communities at the timberlines of Sejila Mountain.”
to
“In the present study, soil fungal diversity, community structure and ecological function of the north and south timberlines of the Sejila Mountain in China’s southeastern Tibet along a 200m elevation gradient were explored. The findings showed that the site effect of timberlines and A. georgei presence exerted apparent influences on the soil fungal community structure and function at the two timberlines. These soil fungal community traits were related to coverage and density, soil pH and ammonium nitrogen at the north timberline, while they had no associations with the vegetation and soil factors at the south timberline.”.
Please refer to the conclusion section in the revised manuscript.
Table 3. should be presented in the Results section
Response:
We sincerely appreciate your valuable feedback. We have moved Tables 2 and 3 to page 8 and 10 of the revised manuscript and provided the following descriptions. Please refer to the revised manuscript.
Added text:
3.1. Vegetation and soil characteristics
3.1.1 Vegetation
The two-way ANOVA (Table 2) showed that tree density (F=5.21, P=0.03) at the north timberline was significantly higher than that at the south timberline. Shrub coverage (F=11.72, P<0.01) in the shrublands surpassed that in the forests and forest lines. Moreover, tree coverage (F=111.73, P<0.01) and density (F=25.14, P<0.01) were considerably greater in the forests and lower in the shrublands compared to the other four vegetation zones, respectively. The interaction between timberline site and vegetation zone had a significant effect on shrub coverage (F=3.18, P=0.04). Specifically, at the south timberline, shrub coverage in the forest was lower than that in the other vegetation zones, shrub coverage increased gradually from forest to shrubland. Tree coverage and density, however, were highest in the forest compared to other zones, exhibiting a decline from forest to shrubland. The trend on the north timberline was similar, but shrub coverage in the shrubland was significantly higher than that in the forest and forest line. Nonetheless, tree coverage and density were substantially greater in the forest than in the other vegetation zones.
3.1.2 Soil
The two-way ANOVA (Table 3) indicated that the soil field water holding capacity (F=6.14, P=0.02) at the north timberline was significantly higher than that at the south timberline, while the soil bulk density (F=3.16, P=0.04) in the forests and tree lines was significantly higher than that in the tree species lines. There was also a significant interaction between timberline site and vegetation zone on soil bulk density (F=4.00, P=0.02). Specifically, the soil bulk density in the forest at the south timberline displayed a notably higher value than that in other vegetation zones, while the soil field water holding capacity in the tree species line surpassed both forest and shrubland by a significant margin. At the north timberline, only tree line's soil bulk density displayed a significant elevation from other vegetation zones.
Soil pH (F=45.21, P<0.01) and available phosphorus content (F=27.80, P<0.01) were significantly higher at the south timberline when compared to the north timberline, whereas the soil ammonium nitrogen content (F=13.84, P<0.01) was significantly higher at the north timberline (Table 3). The available phosphorus content (F=3.56, P=0.02) was significantly higher in the forests than in the other three vegetation zones except tree lines. Moreover, the interaction between timberline site and vegetation zone had a significant effect on soil pH (F=16.30, P<0.01), total phosphorus (F=5.04, P=0.01), and ammonium nitrogen (F=3.83, P=0.02). The pH levels in the south timberline's forest and forest line were notably distinct from other vegetation zones, while the soil organic carbon concentration in the shrubland was considerably higher than that in the forest. Furthermore, the total phosphorus content of tree line was significantly greater than shrub. On the other hand, the north timberline exhibited lower soil pH and available phosphorus levels in the forest, as compared to other vegetation zones. Additionally, forest and tree line had a significantly higher soil ammonium nitrogen content than tree species line and shrub.
The soil C/N (F=87.27, P<0.01) was significantly higher at the north timberline than at the south timberline, while shrublands had a significantly higher C/N (F=36.02, P<0.01) than the other vegetation zones (Table 3). Additionally, soil C/N (F=17.68, P<0.01) and C/P (F=3.63, P=0.02) were significantly affected by the interaction between timberline site and vegetation zone. Notably, the C/N and C/P of shrub at the south timberline were significantly higher than those of other vegetation zones, while the N/P of shrub was significantly higher than that of forest and tree line. Finally, the C/N of shrub at the north timberline was significantly higher than that of other vegetation zones except forest line.
Page 7, lines 34-36: move to the Results section
Response:
Thank you very much for your suggestion. We have moved these sentences to the Results section of the revised manuscript. Please refer to the top of page 11 of the revised manuscript.
Reviewer 2 Report
Dear Editor/Author,
The results of the manuscripts are valuable for science, however, I do have some concerns that I believe would benefit from further attention before publication.
Introduction:
Paragraph 2 (Lines 37-61) is too long and contains few messages. It is confusing to understand the message of this paragraph. I would suggest split it in two and make clear message in each paragraph.
Line 66-67: Give an example for the statement
Results:
Page 13: There is a section “3.3. Ecological function of soil fungi” which has not been explained in M & M. How did you recognised the function of fungi? It has been mentioned that some are pathogen. I am not sure about this, how about those are endophyte but under stress condition become pathogen? Also the fungi must be identified at the species level to be able to say if they are pathogen or saprophyte.
Discussion:
Needs to improve with better structure and writing. I found it difficult to understand it, there are many messages and conclusions in each paragraph. I would recommend to follow the bellow general structure for writing of each paragraph:
- Topic sentence for the paragraph
- Outcome of this study
- Comparing with the other studies
- One sentence of conclusion
Author Response
Response to Reviewer 2
The results of the manuscripts are valuable for science, however, I do have some concerns that I believe would benefit from further attention before publication.
Introduction:
Paragraph 2 (Lines 37-61) is too long and contains few messages. It is confusing to understand the message of this paragraph. I would suggest split it in two and make clear message in each paragraph.
Response:
Thank you very much for your feedback. We completely agree with you, the paragraph was indeed too lengthy. We have now divided it into two parts, with the first part mainly describing ectomycorrhizal fungi and the second part focusing on saprotrophic fungi. Please refer to the revised Introduction section for these changes.
Line 66-67: Give an example for the statement
Response:
Thank you very much for your feedback, we completely agree. We have already added sentences with relevant examples to this paragraph. Please see the Introduction section of the revised manuscript.
Results:
Page 13: There is a section “3.3. Ecological function of soil fungi” which has not been explained in M & M. How did you recognised the function of fungi? It has been mentioned that some are pathogen. I am not sure about this, how about those are endophyte but under stress condition become pathogen? Also the fungi must be identified at the species level to be able to say if they are pathogen or saprophyte.
Response:
Thank you very much for your feedback. We have added the following description to the "2.5.3. OTU clustering and taxonomic annotation" of Materials and Methods. Please refer to the revised manuscript for details.
The prediction of fungal function is based on FUNGuild [1], a widely-used practical tool that links the fungal marker gene sequence information obtained from high-throughput sequencing to the ecological functions of fungi [2-8]. "Guild" functional group refers to a group of species that, regardless of their systematic evolution, are classified into the same group because they adopt similar ways of absorbing and utilizing environmental resources. Fungi can be divided into three main categories based on their nutritional mode: Pathotroph, Saprotroph, and Symbiotroph. Pathotroph is further subdivided into Animal Pathogen, Plant Pathogen, Fungal Parasite, Lichen Parasite, Bryophyte Parasite, and Clavicipitaceous Endophyte. Saprotroph is further subdivided into Dung Saprotroph, Leaf Saprotroph, Plant Saprotroph, Soil Saprotroph, and Wood Saprotroph. Symbiotroph is further divided into arbuscular mycorrhizal, Ectomycorrhizal, Ericoid Mycorrhizal, Endophyte, Epiphyte, and Lichenized. All of these can be obtained by inputting OTU annotation information into the platform (http://www.stbates.org/guilds/app.php). Terms like Pathotroph-Saprotroph-Symbiotroph, Pathotroph-Saprotroph, Pathotroph-Symbiotroph, Saprotroph-Symbiotroph, indicate that fungi possess two or three nutritional characteristics simultaneously.
We fully agree with your suggestion that functional classification should be carried out at the species level, but currently there are still a considerable number of OTUs that cannot be annotated to the species level, making it difficult to know their functions. However, as more and more fungi with clear species are added to the database, we believe that it will be easier to identify fungal functions in the future.
Added text:
“FUNGuild is a functional annotation tool used to predict the functional composition of soil fungal communities of fungal amplicons obtained through high-throughput and other methods. However, it can only annotate information of fungal species on the trophic mode levels (pathotroph, symbiotroph, and saprotroph). The modes can be further subdivided into multiple guides. Each guide comprises the species with similar absorption and utilization of environmental resources [45]. Herein, the ecological function of soil fungi was described based on the mode and guild levels.”
Reference
- Nguyen, H.N.; Song, Z.W.; Bates, T.S.; Branco, S.; Tedersoo, L.; Menke J.; Schilling, J.S.; Kennedy, P.G. Funguild: an open annotation tool for parsing fungal community datasets by ecological guild. Fungal Ecol. 2016, 20, 241–248.
- Gravuer, K.; Eskelinen, A.; Winbourne, J.B.; Harrison, S.P. Vulnerability and resistance in the spatial heterogeneity of soil microbial communities under resource additions. Proc. Natl. Acad. Sci. USA. 2020, 117, 7263-7270.
- Wu, L.; Zhang, Y.; Guo, X.; Ning, D.; Zhou, X.; Feng, J.; Yuan, M.M.; Liu, S.; Guo, J.; Gao, Z.; Ma, J.; Kuang, J.; Jian, S.; Han, S.; Yang, Z.; Ouyang, Y.; Fu, Y.; Xiao, N.; Liu, X.; Wu, L.; Zhou, A.; Yang, Y.; Tiedje, J.M.; Zhou, J. Reduction of microbial diversity in grassland soil is driven by long-term climate warming. Nat Microbiol. 2022, 7, 1054-1062.
- Bello, A.; Wang, B.; Zhao, Y.; Yang, W.; Ogundeji, A.; Deng, L.; Egbeagu, U.U.; Yu, S.; Zhao, L.; Li, D.; Xu, X. Composted biochar affects structural dynamics, function and co-occurrence network patterns of fungi community. Sci Total Environ. 2021, 775, 145672.
- Zarza, E.; López-Pastrana, A.; Damon, A.; Guillén-Navarro, K.; García-Fajardo, L.V. Fungal diversity in shade-coffee plantations in Soconusco, Mexico. PeerJ. 2022, 10, e13610.
- Xiang, L.G.; Wang, H.C.; Wang, F.; Cai, L.T.; Li, W.H.; Hsiang, T.; Yu, Z.H. Analysis of phyllosphere microorganisms and potential pathogens of tobacco leaves. Front Microbiol. 2022, 13, 843389.
- Villalobos-Flores, L.E.; Espinosa-Torres, S.D.; Hernández-Quiroz, F.; Piña-Escobedo, A.; Cruz-Narváez, Y.; Velázquez-Escobar, F.; Süssmuth, R.; García-Mena, J. The bacterial and fungal microbiota of the Mexican Rubiaceae Family medicinal plant Bouvardia ternifolia. Microb Ecol. 2022, 84, 510-526.
- Sun, R.; Yi, Z.; Fu, Y.; Liu, H. Dynamic changes in rhizosphere fungi in different developmental stages of wheat in a confined and isolated environment. Appl Microbiol Biotechnol. 2022, 106, 441-453.
Discussion:
Needs to improve with better structure and writing. I found it difficult to understand it, there are many messages and conclusions in each paragraph. I would recommend to follow the bellow general structure for writing of each paragraph:
- Topic sentence for the paragraph
- Outcome of this study
- Comparing with the other studies
- One sentence of conclusion
Response:
Thank you very much for your suggestion, we completely agree. We have made modifications to the discussion section according to the general structure you proposed. Please refer to the discussion section of the revised manuscript.
Reviewer 3 Report
The work presented in this paper deals with the characterization of the diversity and ecological function of soil fungal communities at the timberline transition zones of a Tibetan mountain according to the north or south exposure of the slopes. With the growing concern on the global climate change, the choice of this altitudinal transition zone is particularly welcome.
This article is clear and well written. The sampling pattern is excellent. The methods used for the DNA analysis of the fungal community are classical and carefully applied. The discussion is based on a sound analysis of the results.
I do not see any flaw that could delay the publication of this very good article.
Author Response
Response to Reviewer 3
Thank you very much for your evaluation of this manuscript, which is a great encouragement to us. In order to further improve the quality of the paper, we have made further revisions to the manuscript. Please refer to the revised manuscript. Thank you again for taking the time to read the manuscript.